# Genetic transformation of *Spizellomyces punctatus*, a resource for studying chytrid biology and evolutionary cell biology

Edgar M Medina[1,2], Kristyn A Robinson[3], Kimberly Bellingham-Johnstun[4], Giuseppe Ianiri[2], Caroline Laplante[4], Lillian K Fritz-Laylin[3], Nicolas E Buchler[4]*

[1]University of Program in Genetics and Genomics, Duke University, Durham, United States; [2]Department of Molecular Genetics and Microbiology, Duke University, Durham, United States; [3]Department of Biology, University of Massachusetts, Amherst, United States; [4]Department of Molecular Biomedical Sciences, North Carolina State University, Raleigh, United States

**Abstract** Chytrids are early-diverging fungi that share features with animals that have been lost in most other fungi. They hold promise as a system to study fungal and animal evolution, but we lack genetic tools for hypothesis testing. Here, we generated transgenic lines of the chytrid *Spizellomyces punctatus*, and used fluorescence microscopy to explore chytrid cell biology and development during its life cycle. We show that the chytrid undergoes multiple rounds of synchronous nuclear division, followed by cellularization, to create and release many daughter 'zoospores'. The zoospores, akin to animal cells, crawl using actin-mediated cell migration. After forming a cell wall, polymerized actin reorganizes into fungal-like cortical patches and cables that extend into hyphal-like structures. Actin perinuclear shells form each cell cycle and polygonal territories emerge during cellularization. This work makes *Spizellomyces* a genetically tractable model for comparative cell biology and understanding the evolution of fungi and early eukaryotes.

**\*For correspondence:**
nebuchle@ncsu.edu

**Competing interests:** The authors declare that no competing interests exist.

## Introduction

Zoosporic fungi, commonly referred to as 'chytrids', span some of the deepest fungal Phyla and comprise much of the undescribed environmental fungal DNA diversity in aquatic ecosystems (*James et al., 2006*; *Richards et al., 2012*; *Powell and Letcher, 2014*; *Grossart et al., 2016*). Many chytrids are saprophytes or parasites of photosynthetic organisms and actively shuttle carbon to higher trophic levels (*Kagami et al., 2014*; *Grossart et al., 2016*). Other chytrids are animal parasites, including the *Batrachochytrium* genus that includes the frog-killing *B. dendrobatidis* (*Longcore et al., 1999*) and salamander-killing *B. salamandrivorans* that are devastating global amphibian populations (*Martel et al., 2013*).

Chytrids are unique in that they have retained ancestral cellular features, shared by animal cells and amoebae, while also having fungal features. For example, chytrids begin their life as motile zoospores that lack a cell wall, swim with a single posterior cilium nucleated from a centriole, and crawl across surfaces (*Fuller, 1976*; *Sparrow, 1960*; *Deacon and Saxena, 1997*; *Held, 1975*; *Fritz-Laylin et al., 2017b*). Later life cycle stages exhibit fungal characteristics including the formation of chitinous cell walls, the growth of hyphal-like structures, and the development of a sporangium (sporangiogenesis); see *Figure 1*. Chytrid zoosporogenesis involves multiple rounds of mitosis without cytokinesis to create a multi-nuclear coenocyte, followed by cellularization to form zoospores with a single nucleus. The formation of a multinuclear compartment followed by cellularization is reminiscent of development in flies (e.g. *Drosophila*), amoeba (e.g. *Physarum*), and protozoa (e.g. *Plasmodium*). Although there are important differences from fly embryos (particularly the need for

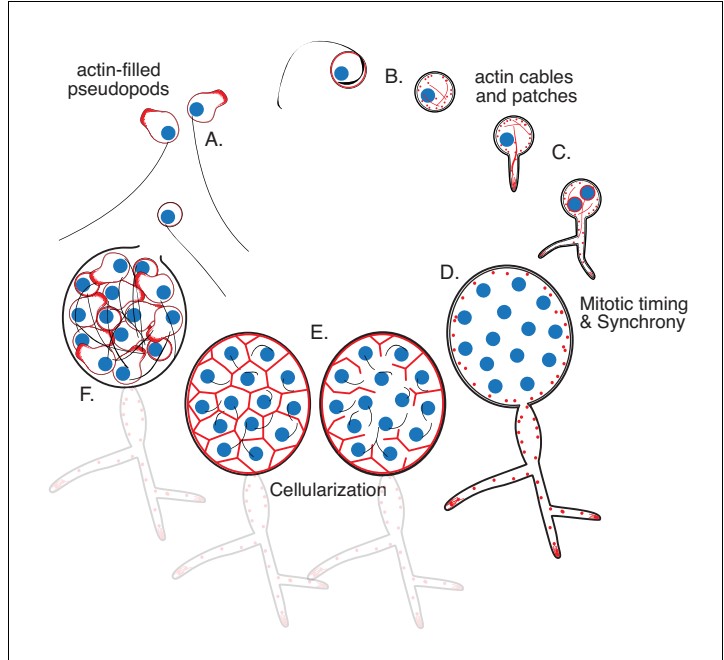

**Figure 1.** Life cycle of the chytrid *Spizellomyces punctatus*. Timeline and events as measured in this work. The chytrid produces globular zoospores (3–5 µm) that swim with a motile cilium (20–24 µm). (A) The uninucleate zoospore (nucleus in blue) has a cilium associated with a basal body. Swimming zoospores can also crawl on surfaces using amoeboid-like motion (polymerized actin in red). (B) The start of encystment (before 1 hr) occurs when the cilium retracts by a lash-around mechanism, followed by formation of cell wall (*Koch, 1968*). (C) The cyst then germinates and forms a single germ tube (at 1–3 hr) that later expands and branches into a rhizoidal system. The nucleus remains in the cyst during germ tube expansion as the cyst develops into a single reproductive structure called the sporangium. The first mitotic event (at 8–12 hr) usually correlates with the ramification of rhizoids from the germ tube. (D) Mitosis in the sporangium is coordinated with growth, as nuclei replicate and divide in a shared compartment. There can be a total of five to eight synchronous mitotic cycles as each sporangium develops a branched rhizoid system with subsporangial swelling in the main rhizoid. (E) Mitosis halts and zoospore formation begins in the sporangium. Ciliogenesis likely occurs before cellularization as in other chytrids (*Renaud and Swift, 1964*). (F) The nuclei cellularize and develop into zoospores while the sporangium develops discharge papillae. Once cellularization is complete and environmental conditions are favorable, the zoospores will escape the sporangium through the discharge papillae (at 20–30 hr). Diagram not drawn to scale. Times are relative to the start of microscopy after zoospore harvest.

the chytrid sporangium to extract nutrients from the environment and coordinate growth with the cell cycle), determining the mechanisms controlling chytrid cellularization provides a comparative framework for understanding cellularization in animals and other eukaryotic lineages.

The major bottleneck to studying chytrids in molecular detail has been the lack of a model organism with tools for genetic transformation. Here, we describe the successful adaptation of *Agrobacterium*-mediated transformation to generate reliable and stable genetic transformation of the soil chytrid *Spizellomyces punctatus*. We expressed fluorescent proteins fused to histone and actin-binding proteins to characterize the development and cell biology of *Spizellomyces* throughout its life cycle using live-cell imaging. Below, we will show that *Spizellomyces* is a well-suited model system for uncovering molecular mechanisms of cell cycle regulation, cell motility, and development because it is fast-growing, displays both crawling and swimming motility, and possesses a characteristic chytrid developmental life cycle. These tools allow, for the first time, direct molecular probing to test new hypotheses about the evolution and regulation of the cell cycle (*Medina et al., 2016*), cell motility (*Fritz-Laylin et al., 2017b*), and development in chytrid fungi.

## Results

### Developing tools for genetic transformation

The plant pathogen *Agrobacterium tumefaciens* normally induces plant tumors by injecting and integrating a segment of transfer DNA (T-DNA) from a tumor-inducing plasmid (Ti-plasmid) into the plant genome. Researchers have exploited this feature to integrate foreign genes in plants by cloning them into the T-DNA region of the Ti-plasmid, inducing virulence genes for processing/transport of T-DNA, and co-culturing induced *Agrobacterium* with the desired plant strain. Because *Agrobacterium*-mediated transformation has been adapted for transformation of diverse animals and fungi (*Bundock et al., 1995*; *Kunik et al., 2001*; *Covert et al., 2001*; *Ianiri et al., 2017*; *Vieira and Camilo, 2011*), we chose to use this system in *Spizellomyces punctatus*. To this end, we modified an *Agrobacterium* plasmid to integrate and express a selectable marker (e.g. drug resistance) in *Spizellomyces*.

To determine a suitable selection marker for *Spizellomyces*, we tested the effects of drugs on the growth of the chytrid on agar plates. We spread zoospores on plates with various concentrations of drugs and assessed the cultures for cell growth, colony formation, and zoospore release using light microscopy. Although Geneticin (G418), Puromycin, and Phleomycin D10 (Zeocin) did not inhibit growth up to 800 mg/L, we determined that 200 mg/L Hygromycin and 800 mg/L Nourseothricin (CloNAT) resulted in complete absence of growth after 6 days of incubation at 30℃. All remaining experiments were performed using Hygromycin (200 mg/L).

Next, we identified *Spizellomyces* promoters that can drive gene expression at sufficient levels to provide resistance to Hygromycin and measurable protein fluorescence. In the absence of a chytrid system to perform these tests, we reasoned that *Spizellomyces* promoters that express at high levels in yeast (*Saccharomyces cerevisiae*) would likely also work in chytrids. Therefore, we used an *Agrobacterium* plasmid (*Ianiri et al., 2017*) that propagates in yeast to first screen *Spizellomyces* promoters that successfully express a fusion of Hygromycin resistance (hph) and green fluorescent protein (GFP); see Materials and methods. We confirmed that *Spizellomyces* HSP70 and H2B promoters resulted in resistance to Hygromycin as well as measurable GFP fluorescence in yeast via flow cytometry (*Figure 2—figure supplement 1A*) and microscopy. All remaining experiments were performed using the stronger H2B promoter.

With active promoters and effective selection, we performed *Agrobacterium*-mediated transformation by co-culturing *Spizellomyces* zoospores with *Agrobacterium* carrying H2Bpr-hph-GFP; see Materials and methods. Although Hygromycin-resistant, none of the GFP transformants (*Figure 2—figure supplement 1B*) exhibited green fluorescence above background. This has been seen in other emerging model systems (i.e. choanoflagellate *Salpingoeca Booth et al., 2018*) and is likely due to GFP misfolding. When GFP was replaced by tdTomato, we obtained transformants that exhibited both Hygromycin resistance (*Figure 2—figure supplement 1B*) and cytoplasmic fluorescence (*Figure 2—figure supplement 2*). Further tests with other fluorescent protein fusions showed that mClover3, mCitrine, and mCerulean3 are functional in *Spizellomyces* (*Figure 2—figure supplement 2*). We then designed a construct with greater applicability, in which the selectable marker and fluorescent protein are expressed independently and where the fluorescent protein (tdTomato) is fused in-frame to the C-terminus of a protein of interest (POI). This design exploits the compact and divergent architecture of the *Spizellomyces* H2A/H2B promoters to express (POI)-tdTomato in an upstream direction (H2B promoter) while expressing hph in a downstream direction (H2A promoter). As a proof of concept and because we were interested in following nuclear dynamics to measure the timing and synchrony of mitotic events (e.g. DNA segregation, see next section), our first protein of interest was histone H2B; see *Figure 2A*.

### Mitotic cycles are fast and highly synchronous during sporangiogenesis

Chytrid transformation with H2B-tdTomato resulted in bright nuclear localization of fluorescence when compared to cytoplasmic tdTomato (*Figure 2B*). The presence of the T-DNA (total size of 4280 bp) in the transformants was confirmed using PCR for hph and H2B-tdTomato (*Figure 2—figure supplement 3*) and through Southern blot analysis of hph (*Figure 2C*). The results were consistent with random, single T-DNA genomic integration events in each transformant. To determine the location of the T-DNA integrations, we identified the genomic region adjacent to the left border

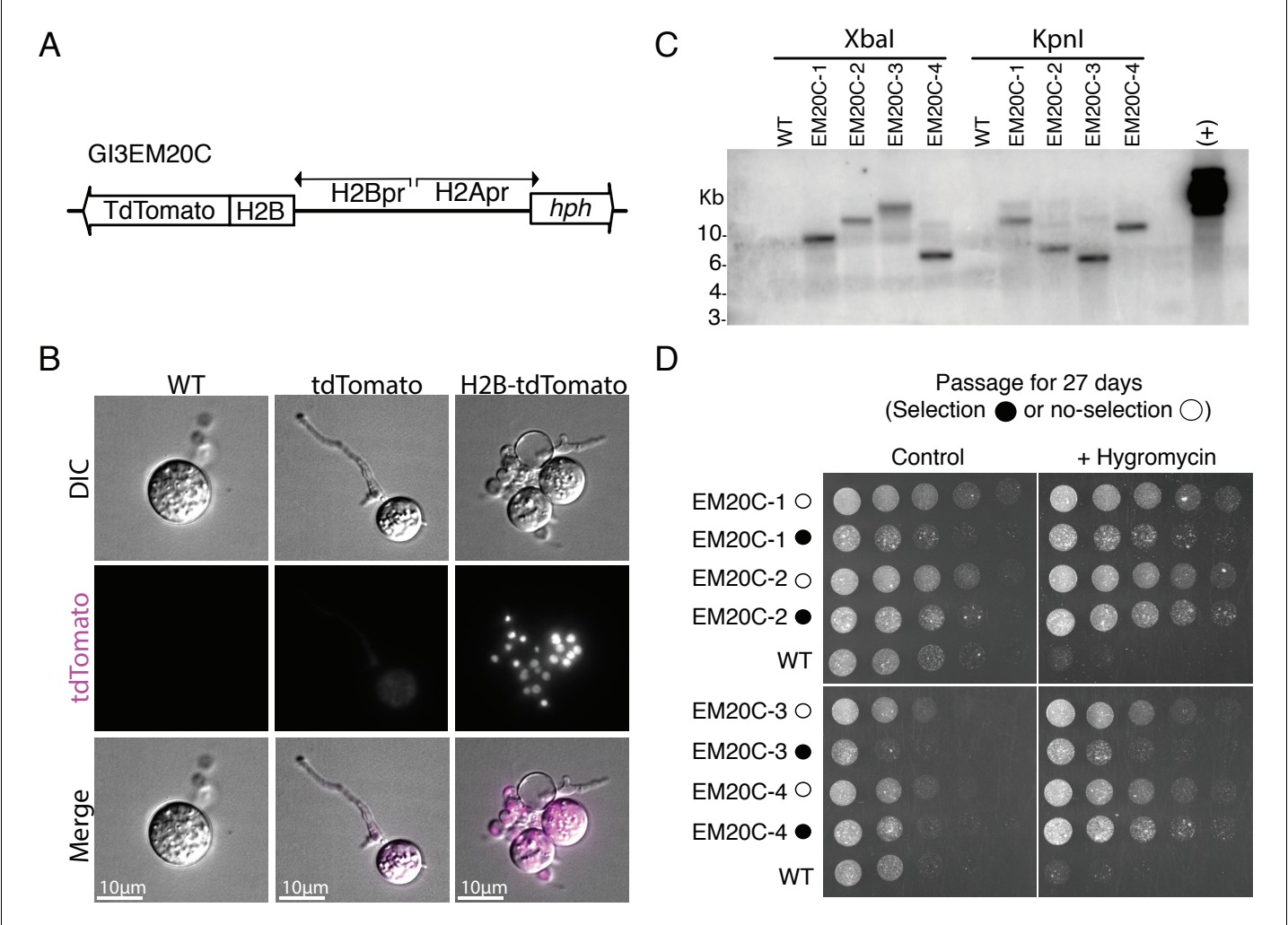

**Figure 2.** Genomic integration of H2B-tdTomato using *Agrobacterium*-mediated transformation. (**A**) Plasmid GI3EM20C takes advantage of the divergent architecture of H2A/H2B to express an H2B-tdTomato fusion in an upstream direction (H2B promoter) while expressing hph in a downstream direction (H2A promoter). (**B**) Representative images from wild type (left), and transformants expressing cytoplasmic hph-tdTomato (plasmid GI3EM18) (center) and nuclear-localized H2B-tdTomato (right). Top row shows DIC and the middle row shows fluorescence microscopy at 561 nm with overlaid images on the bottom row. For comparable results, all strains are presented at the same intensity levels used for H2B-tdTomato fluorescence image. Scale bar indicates ten microns. Image acquisition conditions: POL: transmittance 32%, exposure 0.15 s; TRITC filter, maximal projection, transmittance 32%, Exposure 0.2 s, 0.3 micrometers slice thickness. (**C**) Southern blot of four transformants, in which genomic DNA was digested either with XbaI or KpnI and probed using the Hygromycin resistance gene (hph). We used plasmid GI3EM20C as a positive control (+). (**D**) Four independent transformants were transferred, every two days, in both selective and non-selective medium at 30℃ for a total of 27 days (minimum of 23 life cycles or 116–185 mitotic cycles), followed by a challenge on selective medium. These strains were spotted in a twofold dilution series on non-selective and selective (+Hygromycin) plates, and incubated for 2 days at 30℃.

The online version of this article includes the following video, source data, and figure supplement(s) for figure 2:

**Figure supplement 1.** *Spizellomyces* promoters successfully express fluorescent protein and drug-resistance genes.

**Figure supplement 2.** Diverse fluorescent proteins are functional in *Spizellomyces punctatus*.

**Figure supplement 3.** PCR validation of H2B-tdTomato transformants.

**Figure supplement 4.** Mapping T-DNA genomic insertion sites with inverse PCR.

**Figure supplement 5.** Comparison of development and cell cycle timing between EM20C-1 and EM20C-2.

**Figure supplement 5—source data 1.** Development and cell cycle timing data for EM20C-1 and EM20C-2 used to create *Figure 2—figure supplement 5*.

**Figure 2—video 1.** Time-lapse microscopy in developing H2B-tdTomato sporangia taken over the course of 32 hr with images captured every 2 min.
https://elifesciences.org/articles/52741#fig2video1

(LB) of the T-DNA by inverse PCR (*Figure 2—figure supplement 4*). In three of the four transformants (EM20C-1, 2, 3), the T-DNA LB was located within 200 bp upstream of the transcription start site (TSS) of a gene (SPPG_04375 an M48 peptidase, SPPG_03425 an adenine nucleotide hydrolase, and SPPG_02523 a PHO-like cyclin, respectively). For strain EM20C-3, two different DNA:genome junctions were detected in the 5'UTR of the gene SPPG_02523, suggesting an irregular T-DNA insertion. Last, for strain EM20C-4, the LB T-DNA was inserted 844 bp from the closest TSS (SPPG_08788 a hypothetical protein). As observed in *Arabidopsis*, we might expect variation in gene expression based on the genomic locus of integration of the T-DNA fragment. To quantify this variation, we measured H2B-tdTomato expression in our EM20C transformants using flow cytometry (*Figure 2—figure supplement 4*). The data show that strains EM20C-2, 3, 4 showed similar and unimodal levels of tdTomato fluorescence at the population level, despite the different sites of genomic integration. The exception is EM20C-1, which exhibits bimodal gene expression: the top mode is identical to the other transformants, but the bottom mode is half the intensity. Last, we established that the transformants had transgenerational stability by passaging them in non-selective medium for several weeks, followed by a challenge in selective medium (*Figure 2D*).

To quantify the timing and synchrony of the *Spizellomyces* cell cycle, we used live cell epi-fluorescence imaging of H2B-tdTomato strains EM20C-1 and EM20C-2 at 2 min intervals (*Figure 2—figure supplement 5*). Our results show that zoospores have a single nucleus, they retract their flagellum and encyst in less than 1 hr, the germ tube emerges at ~1−3 hr, the first mitotic event (i.e. one nucleus to two nuclei) occurs at ~8−12 hr, and sporangia develop and undergo 5−8 mitotic cycles in less than 30 hr before completing their life cycles and releasing 32–256 zoospores; see *Figure 2—video 1*. To measure all nuclei within a sporangium with better temporal and z-resolution, we followed nuclear dynamics of EM20C-1 at 1 min time intervals using live-cell confocal microscopy of a H2B-tdTomato strain (*Figure 3—video 1*). We measured the number of nuclei over time per sporangium (*Figure 3A*) to estimate the synchrony of nuclear division waves and the period of time between waves of nuclear division. Wave time ($\Delta t$) is the time for a wave of synchronous nuclear divisions to propagate across the sporangium. The cell cycle period ($\tau$) is the interval of time between nuclear division waves. We found that the average cell cycle period was ~150 min and that each wave of nuclear division was completed within 1 min (*Figure 3B*). In addition, by following the compaction and localization of H2B-tdTomato we show that all measurable mitotic events occurred within 5 min, or less than 3.3% of the cell cycle period (*Figure 3B and C*). Altogether, these results show that *Spizellomyces* is mitotically inactive in its early life cycle (zoospore and germination stages) but, once committed, the cell cycle is fast and nuclear divisions within each sporangium are highly synchronous; see *Table 1*.

## Actin polymerization drives zoospore motility

Like their pre-fungal ancestors, chytrids swim with a motile cilium. Some chytrids can also crawl across and between solid substrates, much like amoeba and animal immune cells (*Fuller, 1976*; *Sparrow, 1960*; *Whisler et al., 1975*; *Couch, 1945*; *Deacon and Saxena, 1997*; *Held, 1973*; *Dorward and Powell, 1983*; *Fritz-Laylin et al., 2017b*). Eukaryotes employ multiple strategies to crawl (filopodia, pseudopodia and blebs) that depend on distinct molecular mechanisms (*Fritz-Laylin et al., 2017a*). One form of crawling, the pseudopod-based $\alpha$-motility, relies on the expansion of branched-actin filament networks that are assembled by the Arp2/3 complex and allow cells to navigate complex environments at speeds exceeding 20 µm/min. The activators of branched-actin assembly WASP and SCAR/WAVE have been described as a molecular signature of the capacity for $\alpha$-motility (*Fritz-Laylin et al., 2017b*). *Spizellomyces* has homologs of WASP (SPPG_00537), SCAR/WAVE (SPPG_02302), and its zoospores are proficient crawlers.

To test whether *Spizellomyces* zoospores crawl using $\alpha$-motility, we expressed a LifeAct-tdTomato fusion. LifeAct is a 17 amino acid peptide that binds specifically to polymerized actin in a wide variety of cell types, such as actin patches and cables in yeast and actin-filled protrusions of crawling animal cells (*Riedl et al., 2008*; *Belin et al., 2014*). To confirm that our LifeAct-tdTomato fusion binds specifically to polymerized actin in *Spizellomyces*, we first fixed and stained actin in zoospores (*Figure 4*) and sporangia (*Figure 5*) with fluorescent phalloidin. We then compared the fluorescent images of cells expressing LifeAct-tdTomato and those expressing hph-tdTomato (negative control) relative to phalloidin-staining.

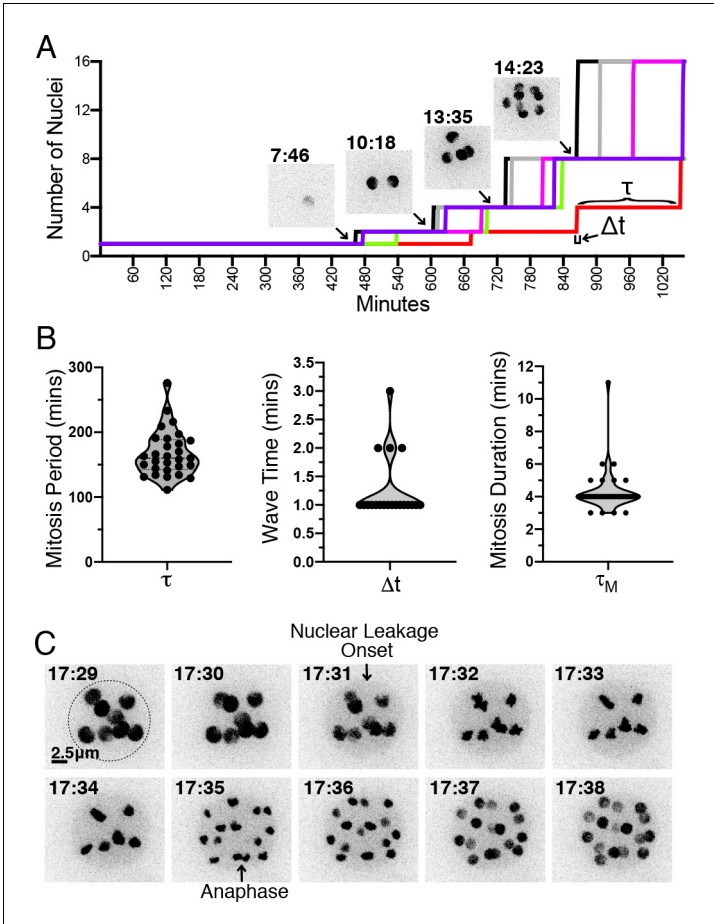

**Figure 3.** H2B-tdTomato reveals the timing and synchrony of mitotic events during sporangiogenesis. (A) Number of nuclei as a function of time during the development of a sporangium, along with H2B-tdTomato fluorescence images from select time points. Each colored line corresponds to a different sporangium. The cell cycle period ($\tau$) is the interval of time between waves of nuclear division (i.e. metaphase to anaphase transition). The wave time ($\Delta t$) is the interval of time for a synchronous wave of nuclear division to sweep across the sporangium. (B) Distribution of cell cycle period ($\tau$), wave time ($\Delta t$) and duration of mitosis ($\tau_M$) across multiple cell cycles. (C) Timing of mitotic events. H2B-tdTomato permits observation of (1) leakage from the nucleus likely due to fenestration of nuclear envelope by the mitotic spindle (**Heath, 1980**; **Fuller, 1976**), followed by chromosome condensation, and (2) chromosome separation during anaphase. Dotted line highlights the cell wall of the sporangium. This particular example shows a mitosis duration of 4 min ($\tau_M$=time from nuclear leakage to anaphase). Time in hr:min. Scale 2.5 micrometers. Distributions are from one time-lapse movie of EM20C-1 (6 cells).

The online version of this article includes the following video and source data for figure 3:

**Source data 1.** Number of nuclei per cell as a function of time (min).

**Figure 3—video 1.** Time-lapse microscopy of nuclear divisions in developing H2B-tdTomato sporangia taken over the course of 24 hr with images captured every 1 min.

https://elifesciences.org/articles/52741#fig3video1

In contrast to the relatively homogeneous distribution of fluorescence in hph-tdTomato zoospores, both fixed (*Figure 4A–C*) and living zoospores (*Figure 4D–E*, and *Figure 4—video 1*) of the LifeAct-tdTomato strain showed a thin layer of fluorescence at the cell cortex, and high levels of fluorescence in the pseudopods at the leading edge. Because the LifeAct fusion localization nearly perfectly correlated with phalloidin intensities in fixed cells, but the hph fusion did not, we presume that this fluorescence represents polymerized actin. The minor deviations between Lifeact and phalloidin can be explained by intrinsic biases of the probes. All live-cell probes of filamentous actin (e.g. Lifeact, F-tractin, Utrophin actin-binding domain) have different biases in their patterns of F-actin

**Table 1.** Comparison of nuclear division synchrony for different coenocytic organisms.

Wave time ($\Delta t$) is defined as the average interval of time for a wave of synchronized nuclear divisions to propagate across the coenocytic nuclei. The nuclear division period ($\tau$) is the average interval of time between waves. Organisms are listed from highest to lowest synchrony index.

| Coenocytic organism | Length scale | Wave time | Wave speed | Nuclear division | Synchrony index | References |
|---|---|---|---|---|---|---|
| | (μm) | $\Delta t$ (min) | (μm min$^{-1}$) | Period $\tau$ (min) | $100\% \cdot (1 - \Delta t / \tau)$ | |
| *Physarum polycephalum* (amoeba) | 1000 | 2 | 500 | 840 | 99.8% | *Halvorsrud et al., 1995* |
| *Spizellomyces punctatus* (fungi) | 5–10 | 1 | 5–10 | 150 | 99.3% | This work |
| *Creolimax fragrantissima* (holozoa) | | 20 | | 300 | 93.3% | *Suga and Ruiz-Trillo, 2013* |
| *Drosophila melanogaster* (metazoa) | 500 | 1.5 | 360 | 10 | 85.5% | *Deneke et al., 2016* |
| *Aspergillus nidulans* (fungi) | 700 | 20 | 35 | 60 | 66.7% | *Clutterbuck, 1970*; *Momany and Taylor, 2000* |

localization and dynamics (*Belin et al., 2014*). None of them can fully recapitulate the patterns seen with phalloidin, but they do provide new insights into the live cell dynamics of actin. The distribution and dynamics of actin that we see in zoospores is in agreement with $\alpha$-motility and actin localization observed for fixed cells during zoospore crawling of *Neocallimastix* (*Li and Heath, 1994*) and *Batrachochytrium* (*Fritz-Laylin et al., 2017b*).

### Actin polymerization during sporangiogenesis

Once the chytrid zoospore encysts, it builds a sporangium by growing both radially and in polarized fashion during germ tube extension and rhizoid formation (*Figure 1*). During early sporangiogenesis, the nuclei are very dynamic while replicating and dividing but then slow down in late sporangiogenesis, presumably during cellularization and zoospore formation (*Figure 2—video1*). Actin has been reported to play fundamental roles during cellularization in another chytrid, *Allomyces macrogynus* (*Lowry et al., 1998*; *Lowry et al., 2004*). Thus, we expect polymerized actin to play a role in the nuclear dynamics and cellularization during sporangiogenesis in *Spizellomyces*.

Our experiments revealed an actin cytoskeleton distributed primarily between cortical patches and linear structures that resemble the actin cables of other fungal species (*Figure 5A*). Each of these structures was visible in both fixed phalloidin stained cells and living cells expressing the Life-Act-tdTomato gene. We also detected transient perinuclear polymerized actin shells with a period similar to mitotic events (*Figure 5B* and *Figure 5—video 1*), which suggests a role for polymerized actin with nuclei during the cell cycle. In late sporangiogenesis, polymerized actin delineated polygonal zoospore territories (*Figure 5B–C*). This is reminiscent of epithelial cellularization seen during *Drosophila* embryogenesis when syncytial nuclei are encapsulated by cell membrane.

## Discussion

Here, we report stable and robust genetic transformation of the chytrid *Spizellomyces punctatus*. We identified and tested native *Spizellomyces* promoters, including a divergent H2A/H2B promoter that can simultaneously express Hygromycin resistance (hph) and a gene of interest throughout the chytrid life cycle. This design was used to express fluorescent proteins and resistance genes in Dikaryotic marine fungi (data not shown), which suggests they should be useful for transforming other chytrids, such as the pathogenic *Batrachochytrium*. *Agrobacterium* was previously used to transform the aquatic chytrid *Blastocladiella emersonii* (*Vieira and Camilo, 2011*). Unfortunately, genetic transformation of *Blastocladiella* displayed a narrow window of gene expression and could only be detected at the zoosporic stage. The advances described in this study should be useful for random and possibly targeted gene integration in other chytrids. For example, *Agrobacterium* T-DNA is often randomly integrated into the host chromosome as a single copy and, if inserted within a gene, it will disrupt gene function and act as a mutagen (*Gelvin, 2017*). Random insertional mutagenesis by *Agrobacterium* has been exploited for forward genetics in plants and fungi (*Alonso et al., 2003*; *Michielse et al., 2005*; *Idnurm et al., 2017*). In Fungi, *Agrobacterium*-mediated transformation has also been used for targeted gene knock-outs by including 1–1.5 kb long,

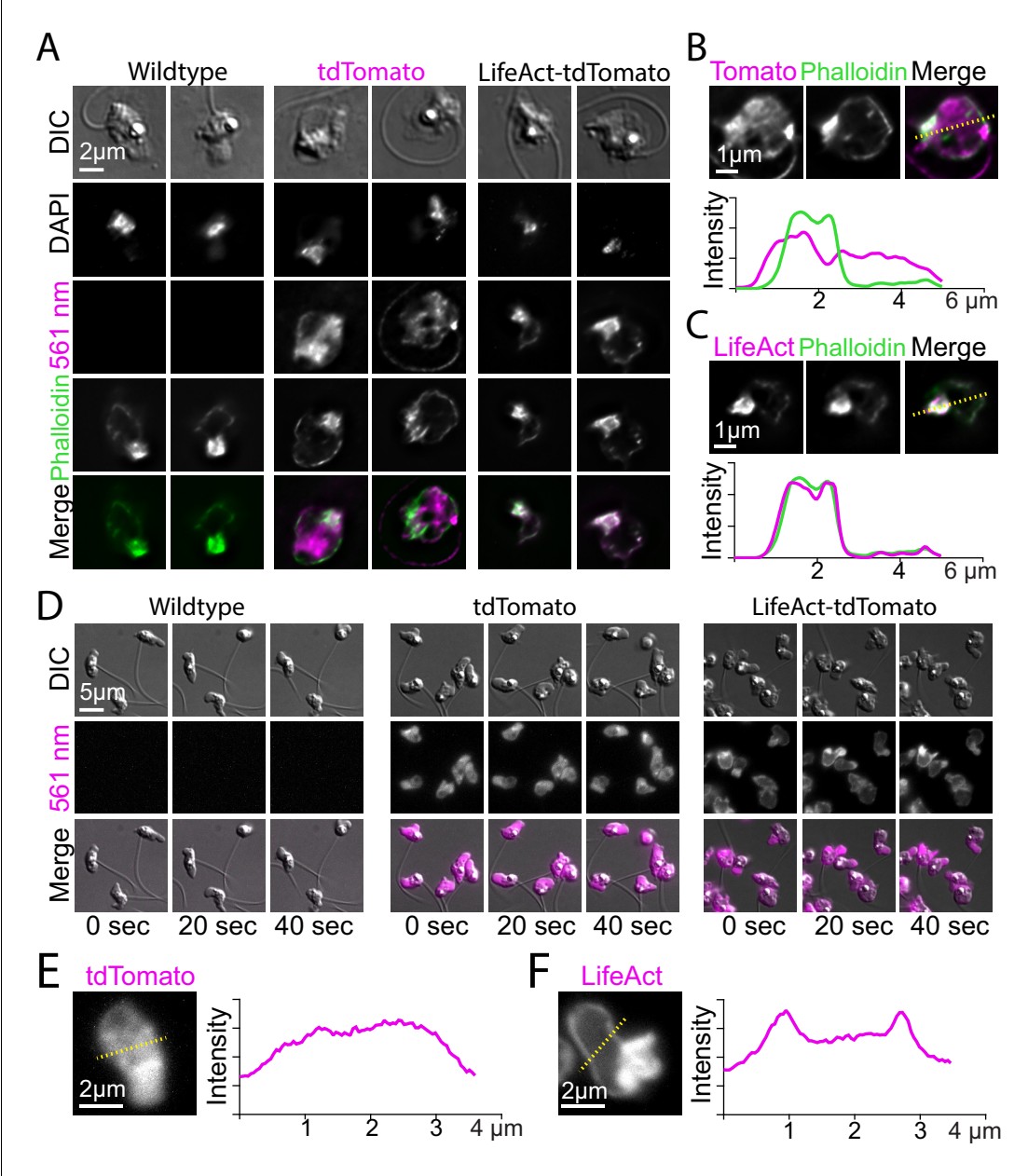

**Figure 4.** Localization of LifeAct-tdTomato in zoospores highlights cortical and pseudopod actin networks. (A) Zoospores from wild type (left), and transformants expressing hph-tdTomato (center) and LifeAct-tdTomato (right) were fixed and stained with fluorescent phalloidin (green). Top row shows DIC and second row shows DNA stain (DAPI). The bottom row shows the phalloidin stain and 561 nm images overlaid. Scale bar indicates two microns. (B and C) Line scan of fixed and stained hph-tdTomato (B) and LifeAct-tdTomato fusion (C). The plot shows line scans of normalized fluorescence intensity of the respective fusion protein (magenta) and fluorescent phalloidin (green). The location for generating the line scans is shown by a yellow dotted line in the image above each plot. Scale bars indicate 1 μm. (D) Stills taken at 20-s intervals from timelapse microscopy of crawling zoospores from the indicated strains at the given timepoints. Images were taken using DIC microscopy (top) and 561 nm fluorescence microscopy (middle), also shown with images merged (bottom). Scale bar indicates 5 μm. E+F) Line scan of fixed and stained hph-tdTomato (E) and LifeAct-tdTomato fusion (F). The plot shows line scans of normalized fluorescence intensity of the respective fusion protein (magenta) and fluorescent phalloidin (green). The location for generating the line scans is shown by a yellow dotted line in the image above each plot. Scale bars indicate 2 μm.

The online version of this article includes the following video for figure 4:

**Figure 4—video 1.** Time-lapse microscopy of crawling wild type, cytoplasmic tdTomato and LifeAct-tdTomato zoospores taken over the course of 2 min with images captured every second, playback in real time.

https://elifesciences.org/articles/52741#fig4video1

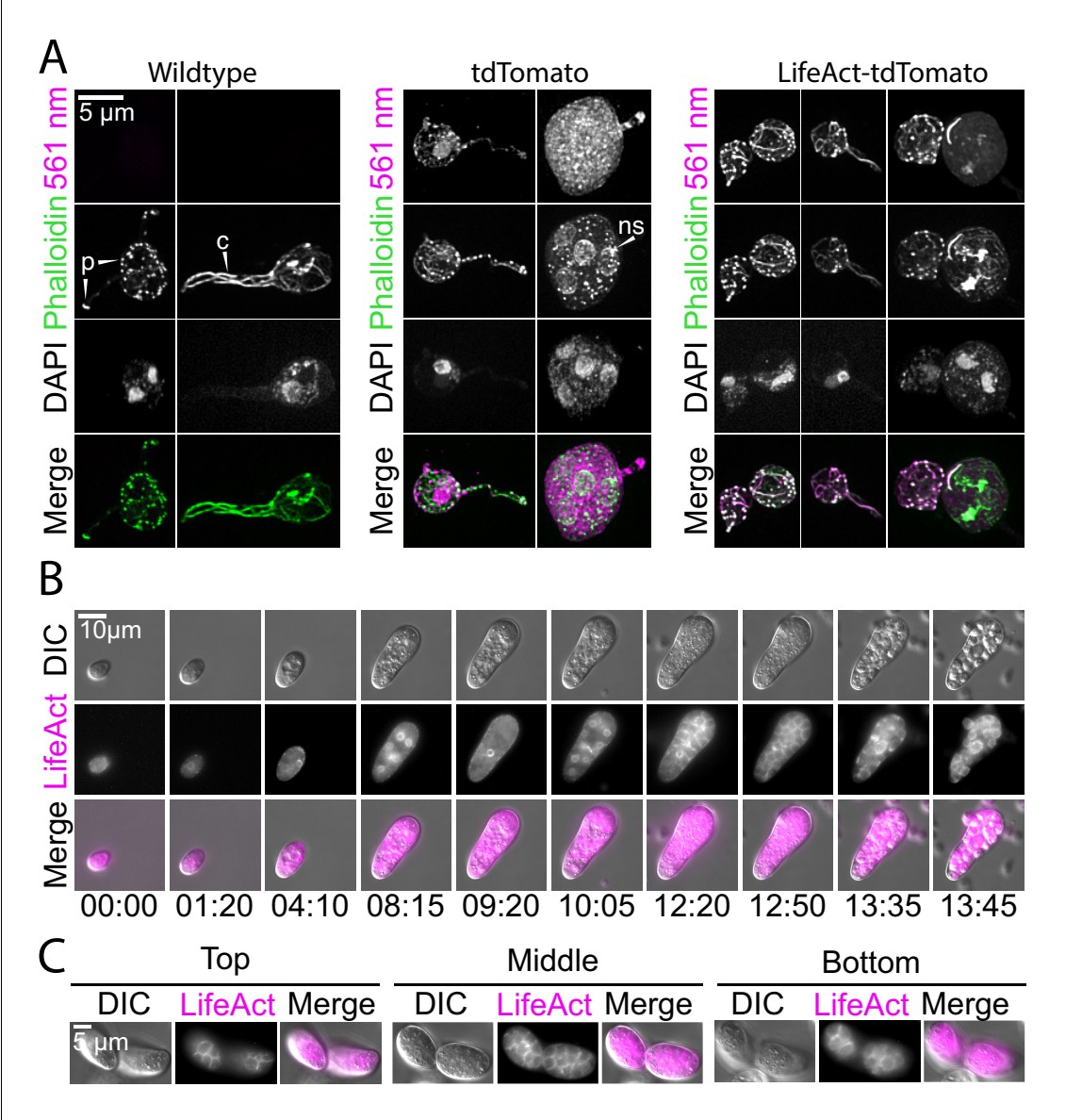

**Figure 5.** Localization of LifeAct-tdTomato in sporangia highlights actin patches, cables, and perinuclear shells. (**A**) Sporangia from wild type (left), and transformants expressing hph-tdTomato (center) and LifeAct-tdTomato (right) were fixed and stained with fluorescent phalloidin (green). The third row shows DNA stain (DAPI). The bottom row shows the phalloidin stain and 561 nm images overlaid. Scale bar indicates 5 μm. Arrows point to examples of actin patches present in sporangia and rhizoids ('p'), cables ('c') and perinuclear actin shells ('ns'). (**B**) Selected stills taken from timelapse microscopy of developing sporangia from the LifeAct-tdTomato transformant strain at times indicated (hr: min). Formation of polygonal territories precedes cellularization. Images taken using DIC (top) and 561 nm (middle), also shown with images merged (bottom). Scale bar indicates ten microns. (**C**) Multiple planes of a single sporangium show how polygonal territories formed during later stages of cellularization encompass the entire cytoplasm.

The online version of this article includes the following video for figure 5:

**Figure 5—video 1.** Time-lapse microscopy of developing LifeAct-tdTomato sporangia taken over the course of 19 hr with images captured every 5 min.

https://elifesciences.org/articles/52741#fig5video1

flanking region of homology in the T-DNA to the desired target gene (*Frandsen et al., 2012*). The feasibility of this approach in a chytrid relies on the rate of homologous recombination, which is in general high in the majority of Fungi.

As a first step to characterize the chytrid cell cycle, we used an H2B-tdTomato fusion and live-cell microscopy to measure the timing of nuclear division. The *Spizellomyces* developmental program

allocates a narrow window (~5 min) of time to the mitotic process during each cell cycle with a highly synchronous wave of nuclear divisions. The level of synchrony is similar to plasmodial nuclei in the amoeba *Physarum polycephalum* or syncytial nuclei during *Drosophila* development, where all nuclei divide within 2 min (*Deneke et al., 2016*) (see *Table 1*). These mitotic dynamics make *Spizellomyces* an interesting comparative model for exploring the physical and molecular determinants of cell division synchrony and its evolution in fungi, animals, and amoeba.

By visualizing actin dynamics throughout the chytrid life cycle, we found that *Spizellomyces* zoospores assemble thick actin cortices and build actin-filled protrusions during motility, similar to other chytrid species (*Fritz-Laylin et al., 2017b*), animal cells, and amoebae. Once the zoospores encyst, there is a drastic shift in actin cytoskeleton organization, in which the cortical shell of polymerized actin was replaced by dynamic puncta (actin patches), some of which are associated with actin cables that often extended into the germ tube or rhizoids. This architecture is typical of fungi, where actin patches are associated with endocytosis and cell wall deposition, whereas actin cables are pathways for targeted delivery of exocytic vesicles (*Lichius et al., 2011*). This biphasic actin distribution – an actin cortex and actin-filled protrusions in zoospores, and actin patches and cables in sporangia – indicates that, like its cell cycle regulatory network (*Medina et al., 2016*), the actin cytoskeleton of *Spizellomyces* displays features that resemble those of both animal and fungal cells.

Our results revealed further actin dynamics and organization during the chytrid development. This includes the formation of perinuclear actin shells that are fleetingly detected by live imaging. Previous observations suggested that perinuclear actin shells of anaerobic chytrids were a fixation artifact (*Li and Heath, 1994*); however, our live cell data indicate that these are real, dynamic cellular structures that likely occur in many chytrids. Similar perinuclear actin shells are associated with nuclear lamina in animal cells, where they play a role in changing nuclear shape before and after mitosis (*Clubb and Locke, 1998*; *Clubb and Locke, 1996* or when squeezing through narrow channels (*Thiam et al., 2016*). Although chytrids and other fungi have lost nuclear lamins, it seems likely that *Spizellomyces* perinuclear actin shells are associated with changes in nuclear shape during the cell cycle.

Finally, we showed that polymerized actin is likely involved in the formation of zoospore polygonal territories (*Figure 5B*) before the formation of the cleavage planes during cellularization. In contrast to cellularization in *Drosophila*, which occurs at the surface of the embryo (i.e. two-dimensional cellularization), cellularization in *Spizellomyces* happens in a three-dimensional context within 3 hr. The chytrid cellularization process is reminiscent of the polarized epithelium of the social amoeba *Dictyostelium discoideum* (*Dickinson et al., 2011*; *Dickinson et al., 2012*) and the membrane invagination dynamics during cellularization in the ichthyosporean *Sphaeroforma arctica*, an unicellular relative of animals (*Dudin et al., 2019*). To what extent is the emergence of multicellularity, which appears to have evolved independently in amoeba, animals, and fungi, dependent upon a shared ancestral toolkit? Establishing shared and unique traits between chytrid fungi and other key lineages will provide a powerful cross-lineage experimental system to test core hypotheses on the evolution of multicellularity and derived fungal features. Based on our tools and findings, we expect that *Spizellomyces* will be a useful model system to study the evolution of key animal and fungal traits.

## Materials and methods

### Strains and growth conditions

We used *Spizellomyces punctatus* (Koch type isolate NG-3) Barr (ATCC 48900) for all chytrid experiments. Unless otherwise stated, *Spizellomyces* were grown at 30°C in Koch's K1 medium (1L; 0.6 g peptone, 0.4 g yeast extract, 1.2 g glucose, 15 g agar if plates; *Koch, 1957*). Two days prior to harvesting zoospores, we aliquoted and spread 1 mL of active, liquid culture pregrown in K1 medium onto K1 plates and incubated them to allow zoospores to encyst, mature, and colonize the agar surface. We flooded each active *Spizellomyces* plate with 1 mL of dilute salt (DS) solution (*Machlis, 1953*) and incubated at room temperature. After 1 hr, released zoospores were retrieved by harvesting the DS medium and purified by slowly filtering the harvest in Luer-Lok syringe through an autoclaved syringe filter holder (Advantec REF:43303010) preloaded with a 25 mm Whatman Grade one filter paper (CAT No. 1001–325). Strains listed in Table 3 are available from the Buchler lab upon request.

## Plasmids

We initially tried to transform Spizellomyces via zoospore electroporation using a protocol developed in zoosporic protists, such as *Phytophthora* (*Ah-Fong et al., 2018*). This was unsuccessful and we turned to *Agrobacterium*-mediated transformation because it has worked in other fungi. We used the pGI3 plasmid backbone for *Agrobacterium*-mediated transformation (*Ianiri et al., 2017*), which contains the *Saccharomyces cerevisiae* 2μ replication origin and the *URA3* selectable marker. This allows pGI3 and its derivatives to replicate in *E. coli*, *A. tumefaciens* and *S. cerevisiae*. Complete details of primers and plasmid construction are in *Table 2* and the Appendix. All plasmids are available from Addgene and their RRIDs are listed in the Appendix.

## Agrobacterium-mediated transformation of *Spizellomyces*

We prepared competent *Agrobacterium* EHA105 strains following the protocol of *Weigel and Glazebrook (2006)*. Plasmids were transformed into competent *Agrobacterium* using 0.2 cm cuvettes in a Gene Pulser electroporator (Bio-Rad, USA) at 25 μF, 200 Ω , 2.5kV. Single colonies were streaked on selective plates (Kanamycin). A colony of transformed *Agrobacterium* containing pGI3-derived plasmid was grown overnight at 30°C in 5 mL of Luria-Bertani broth supplemented with Kanamycin (50 mg/L). After centrifugation, the cell pellet was resuspended in 5 mL of IM (*Bundock et al., 1995*), diluted to an $OD_{660}$ of 0.1 and grown under agitation at 30°C until achieving a final $OD_{660}$ of 0.6, at which point the culture was ready for co-culturing with the chytrid (300 μL per transformation). IM is composed of MM salts (*Hooykaas et al., 1979*) and 40M 2-(N-morpholino)ethanesulfonic acid (MES) pH 5.3, 10 mM glucose, 0.5% (w/v) glycerol and 200μM acetosyringone.

In parallel, chytrid zoospores were harvested and pelleted by centrifugation at 800 g for 10 min. Zoospores were then gently resuspended in 300μL of induction medium (IM). We found that one K1 plate provides enough zoospores for one transformation. For every transformation, zoospores and *Agrobacterium* were combined at four different ratios: 1:1, 1:0.25, 0.25:1, and 0.25:0.25 in a total volume of 200μL. To guarantee tight contact between *A. tumefaciens-S. punctatus* cells, the surface of the IM plate was rubbed with the bottom of a sterile glass culture tube to generate slightly concave depressions in each quadrant of the plate, wherein each 200μL co-incubation mixture was spotted. Plates were incubated unsealed for 4 days at room temperature. Mock transformations with empty *Agrobacterium* (no binary plasmid; grown in the absence of plasmid selective medium) were included as a negative control.

After co-incubation, we added 1 mL of DS solution and gently scraped the plate with a razor blade, pooling the different cell ratios into a single 50 mL centrifuge tube, raising the volume to 20 mL with DS solution, and re-suspending clumps by inversion. The mixture was centrifuged at 1000 g for 10 min and the liquid phase was discarded. The remaining pellet was carefully resuspended with DS solution, plated on K1 plates containing Ampicillin (50 mg/L) and Tetracycline (50 mg/L) to select against *Agrobacterium* and Hygromycin (200 mg/L) to select for transformed *Spizellomyces. Spizellomyces* survival controls were performed by plating transformations after co-culture in non-selective K1 media (Ampicillin (50 mg/L), Tetracycline (50 mg/L)). Transformation plates and controls were incubated at 30°C until colonies were observed (5–6 days). All plates were sealed with parafilm to prevent desiccation. Single colony isolates were retrieved with a sterile needle, resuspended in DS solution and re-plated on a selective Hygromycin plate. Chytrid strains listed in *Table 3* are available from the Buchler lab upon request.

## Nucleic acid manipulation

High-molecular-weight genomic DNA extraction was performed using CTAB/Chloroform protocol (CTAB/PVP buffer: 100 mM Tris-HCl pH7.5; 1.4M NaCl; 10 mM EDTA; 1% CTAB; 1% PVP; 1% *β*-Mercaptoethanol (added just before use)). Briefly, 10 plates of the selected strain were grown at 30°C for 2 days and zoospores were harvested and purified as described before. The pellet of zoospores was resuspended directly in 900μL of CTAB/PVP buffer pre-warmed to 65°C and transferred to 2 mL centrifuge tubes. Tubes were centrifuged briefly and incubated at room temperature for 1 hr in a nutating mixer. After 5 min incubation on ice, DNA was extracted with Chloroform (Sigma-Aldrich REF:288306) twice followed by treatment of supernatant with 100 ng RNase A (Biobasic; 60 U/mg Ref:XRB0473) for 30 min at room temperature. DNA was then precipitated by adding 0.2 vol

**Table 2.** Primers used in this study.

Capital letters in SLIC and Gibson primers indicate template binding regions.

| Plasmid creation | Target amplicon (Source DNA) | Primer name | Primer sequence (5'→ 3') |
|---|---|---|---|
| pEM01 | CMV promoter (pAB1T7) | CMV_F | ccctcactaaagggaacaaaagctggagctgagctcGACATTGATTATTGACTAGTTATTAATAG |
| pEM01 | CMV promoter (pAB1T7) | CMV_R | cttttttacccatgttaattaaAGCTCTGCTTATATAGACCTCC |
| pEM01 | hph (pRS306H) | Hyg_F | tatataagcagagctttaattaacATGGGTAAAAAGCCTGAACTC |
| pEM01 | hph (pRS306H) | Hyg_R | tagaagtggcgcgccaTTATTCCTTTGCCCTCGGAC |
| pEM01 | ADH1 terminator (pNB780) | ADH1t_F | agggcaaaggaataatggcgcgccACTTCTAAATAAGCGAATTTCTTATG |
| pEM01 | ADH1 terminator (pNB780) | ADH1t_R | tgacccggcgggggacgaggcaagctaaacaATATTACCCTGTTATCCCTAGC |
| pEM03 | Hsp70 promoter (Spun gDNA) | HSP70_F | actaaagggaacaaaagctggagctgagctcTTTTAAAATCTTGTCTTTGTGC |
| pEM03 | Hsp70 promoter (Spun gDNA) | HSP70_R | ggtgagttcaggcttttttacccatgttaattaaATTGTGCTGATCTTTGGTCC |
| pEM09 | hph (pFA6-GFP(S65T)-hph) | HygR_F | aagatcagcacaatttaattaATGGGTAAAAAGCCTGAACTCAC |
| pEM09 | hph (pFA6-GFP(S65T)-hph) | HygR_R | tcctcctcctcctccTTCCTTTGCCCTCGGACG |
| pEM09 | GFP (pFA6-GFP(S65T)-hph) | GFP_F | gcaaaggaaggaggaggaggaggaggaAGTAAAGGAGAAGAACTTTTCAC |
| pEM09 | GFP (pFA6-GFP(S65T)-hph) | GFP_R | cgcttatttagaagtggcgcgccTATTTGTATAGTTCATCCATGC |
| | Sequencing | pRS-up | aacataggagccggaagcataaagtg |
| | Sequencing | ADHt-dn | ctgccggtagaggtgtggtcaataag |
| pGI3EM9 | Hsp70pr-hph-GFP (pEM09) | GI3EM9IIF | cgttgtaaaacgacggccagtgccaagcttttttaaaatcttgtctttgtgcac |
| pGI3EM9 | Hsp70pr-hph-GFP (pEM09) | GI3EM9IIR | aggaaacagctatgacatgattacgaattcccggtagaggtgtggtcaataag |
| pGI3EM11 | H2B promoter (Spun gDNA) | prH2B_F | cgttgtaaaacgacggccagtgccaagCTTTTATGCTCCAAGCGGAG |
| pGI3EM11 | H2B promoter (Spun gDNA) | prH2B_R | gagttcaggcttttttacccattaattaaTTTGTGTGTGTGATGGATGAG |
| pGI3EM11 | HygR_up | GI3EM11up | ggatcctcctcctccTTCCTTTGCCCTCGGACG |
| pGI3EM18 | tdTomato (pKT356) | HygtdTom_F | cagcactcgtccgagggcaaaggaaggaggaggaggatcc ATGGTGAGCAAGGGCGAG |
| pGI3EM18 | tdTomato (pKT356) | AdhtdTom_R | tcgcttatttagaagtggcgcgcctTTACTTGTACAGCTCGTCCATG |
| pGI3EM20C | H2A/H2B promoter (Spun gDNA) | H2B2D_F | gctttttacccatttaattaatgctgtgtaaggtgtgcg |
| pGI3EM20B/C | H2A/H2B promoter (Spun gDNA) | H2B2D_R | gggtgccatgtcgacttgtgtgtgtgatggatgag |
| pGI3EM20B/C | H2B CDS (Spun cDNA) | H2Bgen_F | atcacacacacaagtcgacatggcacccaaggaagctc |
| pGI3EM20B/C | H2B CDS (Spun cDNA) | H2Bgen_R | gacctcctcgcccttgctcaccatggatccggaggagga tttagcagactggtacttcgtcac |
| pGI3EM20B/C | HygR (pEM03) | AdhHygF | cgttgtaaaacgacggccagtgccaagcttccggtagaggtgtggtcaataag |
| pGI3EM20B/C | HygR (pEM03) | AdhHygR | ccttacacagcattaattaaatgggtaaaaagcctgaactc |
| pGI3EM20B/C | invPCR T-DNA (LB) | 4_LB_R | tgtggaattgtgagcggata |
| pGI3EM20B/C | invPCR/sequencing T-DNA (LB) | ai077_F | agaggcggtttgcgtattgg |
| pGI3EM22C | LifeAct (Synthesized DNA) | TwistLifeact_F | ctataaaaggcgggcgtgt |
| pGI3EM22C | LifeAct (Synthesized DNA) | TwistLifeact_R | gcgcatgaactctttgatga |
| pGI3EM29 | mCitrine (mCitrine-PCNA-19-SV40NLS-4) | EM18Citrine_F | ccgagggcaaaggaaggaggaggaggatcc ATGGTGAGCAAGGGCGAG |
| pGI3EM29 | mCitrine (mCitrine-PCNA-19-SV40NLS-4) | EM18Citrine_R | aagaaattcgcttatttagaagtggcgcgc CTTGTACAGCTCGTCCATGC |
| pGI3EM30 | mClover3 (pKK-mClover3-TEV) | EM18Clover3_F | gagggcaaaggaaggaggaggaggatcc ATGGTGAGCAAGGGCGAG |
| pGI3EM30 | mClover3 (pKK-mClover3-TEV) | EM18Clover3_R | gaaattcgcttatttagaagtggcgcgc CTTGTACAGCTCGTCCATGC |
| pGI3EM31 | mCerulean3 (mCerulean3-N1) | EM18Cerulean3_F | ccgagggcaaaggaaggaggaggaggatcc ATGGTGAGCAAGGGCGAG |

*Table 2 continued on next page*

*Table 2 continued*

| Plasmid creation | Target amplicon (Source DNA) | Primer name | Primer sequence (5′→ 3′) |
|---|---|---|---|
| pGI3EM31 | mCerulean3 (mCerulean3-N1) | EM18Cerulean3_R | aagaaattcgcttatttagaagtggcgcgcc TTACTTGTACAGCTCGTCCATG |
| | mCitrine/Clover/ Cerulean Sequencing | mClover_Down | gtccaagctgagcaaagacc |
| | PCR screen (Set C)/H2Bpr/ Sequencing | H2BprF1 | tttatgctccaagcggagac |
| | PCR screen (Set C)/Sequencing | TomR1 | cttgtacagctcgtccatgc |
| | PCR screen (Set D)/H2Bpr/ Sequencing | H2BprF2 | cgttaaatgacctgctcgaa |
| | PCR screen (Set D)/H2Bpr/ Sequencing | TomR2 | ccatgccgtacaggaacag |
| | PCR screen (Set A)/Southern Blot/Sequencing | HygF1 | gatgtaggagggcgtggata |
| | PCR screen (Set A)/Southern Blot/Sequencing | HygR1 | atttgtgtacgcccgacagt |
| | PCR screen (Set B)/ Sequencing | HygF2 | gtcctgcgggtaaatagctg |
| | PCR screen (Set B)/ Sequencing | HygR2 | cgtctgctgctccatacaag |

of 10M Ammonium acetate and one volume of absolute Isopropanol and incubated at 4°C overnight. DNA pellet was washed with 70% ethanol thrice and resuspended in TE buffer. DNA quality and concentration were determined by gel electrophoresis.

Detection of transgene integration by Southern blot. 1 µg of high-molecular-weight gDNA from each strain was treated overnight at 37°C with 10U of XbaI or KpnI-HF restriction enzymes, resolved on a 1% Agarose 1X TAE (Tris-Acetate-EDTA) gel and blotted to a GE Healthcare Amersham Hybond-N+ membrane. The membrane was hybridized with a 809 bp fragment of the Hygromycin resistance gene amplified with primers HygF1 and HygR1 and radiolabeled with [$\alpha$-$^{32}$P]-dCTP using the Prime-It II Random Primer Labelling Kit (Agilent Technologies; REF:300385) following manufacturer's instructions.

Identification of T-DNA insertion sites by inverse PCR. 2.5 µg of genomic DNA of wild type (WT) and the four transformed strains of *Spizellomyces* (EM20C-1,2,3,4) was digested in a final volume of

**Table 3.** Chytrid strains available from the Buchler lab upon request.
Plasmid column lists *Agrobacterium* plasmids from Appendix 1 used to create chytrid strains. Integrated gene(s) are described using yeast genetic nomenclature.

| Request ID | Strain name | Plasmid | Integrated gene(s) |
|---|---|---|---|
| NBC24 | EM11-1 | pGI3EM11 | *H2Bpr-hph-GFP-ScADH1ter* |
| NBC34 | EM18-1 | pGI3EM18 | *H2Bpr-hph-tdTomato-ScADH1ter* |
| NBC25 | EM20C-1 | pGI3EM20C | *H2Bpr-H2B-tdTomato-ScADH1ter:H2Apr-hph-ScADH1ter* |
| NBC26 | EM20C-2 | pGI3EM20C | *H2Bpr-H2B-tdTomato-ScADH1ter:H2Apr-hph-ScADH1ter* |
| NBC27 | EM20C-3 | pGI3EM20C | *H2Bpr-H2B-tdTomato-ScADH1ter:H2Apr-hph-ScADH1ter* |
| NBC28 | EM20C-4 | pGI3EM20C | *H2Bpr-H2B-tdTomato-ScADH1ter:H2Apr-hph-ScADH1ter* |
| NBC1 | EM22C-1 | pGI3EM22C | *H2Bpr-Lifeact-tdTomato-ScADH1ter:H2Apr-hph-ScADH1ter* |
| NBC71 | EM29-1 | pGI3EM29 | *H2Bpr-hph-mCitrine-ScADH1ter* |
| NBC55 | EM30-1 | pGI3EM30 | *H2Bpr-hph-mClover3-ScADH1ter* |
| NBC44 | EM31-1 | pGI3EM31 | *H2Bpr-hph-mCerulean3-ScADH1ter* |

50μL with 100U of EcoRI-HF (NEB R3101S) or HindIII-HF (NEB R3104S) for 24 hr at 37°C. After assessing the quality of the digestion by gel electrophoresis, the reaction was heat inactivated and 48μL of the digested DNA was incubated with 1μL of T4 ligase (400u/μL) for 48 hr at 4°C. The ligation was purified by chloroform extraction twice, followed by DNA precipitation and resuspended in 30μL of nuclease-free water. 100 ng of ligated product was used for touch-down Inverse PCR reactions in a final volume of 50μL using NEB Phusion high-fidelity DNA polymerase and 3% DMSO following manufacturer instructions and primers ai77_F and 4_LB_R. Touch-down PCR amplification protocol included an initial denaturation step at 98°C for 3 min followed by 10 cycles of amplification in which the annealing temperature was decreased 1C/cycle until an annealing temperature of 62°C was achieved, followed by 20 amplification cycles at 62°C. Annealing and elongation time during all these cycles was 30 s and 6 min, respectively. Amplification was assessed by gel electrophoresis and bands were retrieved using a razor blade, purified using the Promega Wizard SV Gel and PCR Clean-Up System, and Sanger-sequenced using the primer ai77_F.

## Microscopy

H2B-tdTomato-expressing chytrids were harvested from plates, placed in a glass-bottom dish (Mattek), and covered with a 1.5% K1 agarose pad to keep cells healthy and in the plane of focus (*Young et al., 2012*). We harvested zoospores from plates and re-suspended them in Leu/Lys paralyzing solution (*Dill and Fuller, 1971* ) before putting them in glass-bottom dishes, as above. Live-cell epifluorescence was performed on a temperature-controlled Deltavision Elite inverted microscope equipped with 60x/1.42 oil objective and Evolve-512 EMCCD camera using optical axis integration. Optical axis integration (OAI) is a setting on the Deltavision microscope that opens the shutter and continuously excites and measures fluorescence emission while sweeping from top (+20 microns) to bottom of the z-axis (−20 microns) in 0.5 s, integrating directly the intensities onto the CCD chip. OAI has the advantage that total exposure time is reduced relative to a traditional z-stack. Excitation light was 542/27 nm (7 Color InsightSSI) and emission to the camera was filtered by 594/45 nm (TRITC). Epifluorescence live-cell imaging was done at 30°C. Live-cell confocal microscopy was performed on a Nikon Eclipse Ti inverted microscope equipped with a 100x/N.A. 1.49 CFI Apo TIRF oil objective and fitted with a Yokogawa CSU-X1 spinning disk and Andor iXon 897 EMCCD camera. Excitation light was via 488 nm laser. Confocal live-cell imaging was done at 27°C.

LifeAct-expressing zoospores were collected in DS solution and transferred to cover-glass bottom dishes. For phalloidin staining, glass coverslips were plasma cleaned and immediately coated with 0.1% polyethyleneimine for 5 min, washed thrice with water, then overlaid with zoospores or sporangia suspended in DS solution. Cells were allowed to adhere for 5 min before fixation by adding four volumes of 4% paraformaldehyde in 50 mM Cacodylate buffer (pH 7.2). Cells were fixed for 20 min on ice, washed once with PEM buffer (100 mM PIPES, 1 mM EGTA, 0.1 mM MgSO$_4$), permeabilized and stained with 0.1% Triton X in PEM with 1:1000 Alexa Fluor 488 Phalloidin (66 nM in DMSO, Sigma D2660) for 10 min at room temperature, washed with PEM, then mounted onto glass slides using Prolong Gold with DAPI (Invitrogen P36931). Zoospores were imaged on a Nikon Ti2-E inverted microscope equipped with 100x oil PlanApo objective and sCMOS 4mp camera (PCO Panda). Excitation light was via epi fluorescence illuminator at 405 nm, 488 nm, and 561 nm. Sporangia were imaged on a Nikon Ti-E inverted microscope equipped with 100X oil objective and fitted with a Yokogawa X1 spinning disk (CSU-W1) with 50 μm pinholes and Andor xIon EMCCD camera. Excitation light was via 405 nm laser, 488 nm laser, and 561 nm laser. Z-stack fluorescent images were deconvolved with NIS Elements v5.11 using 20 iterations of the Richardson-Lucy algorithm. Image analysis was performed with the ImageJ bundle Fiji (*Schindelin et al., 2012*). All imagings were done at room temperature.

## Flow cytometry

Zoospores from wild-type and EM20C-(1-4) transformants were harvested from plates and put on ice until fluorescence measurement. Flow cytometry was performed on a MACSQuant VYB using the yellow laser (561 nm) to measure forward scatter (FSC), side scatter (SSC), and tdTomato fluorescence (Y2, 615/20 nm filter). The FSC, SSC, Y2 voltage gain settings for each PMT were 350 V, 350 V, and 510 V, respectively. We recorded the pulse height, width, area for a minimum of 100,000

events per strain. Singlets were identified using FSC/SSC width versus height, and a histogram of singlet tdTomato fluorescence (Y2 height) was plotted using the FlowJo software package.

## Acknowledgements

This material is based upon work supported by the National Science Foundation under Grants No. IOS-1827257 (LKF-L) and IOS-1915750 (NEB). Spinning disk confocal microscopy data collection of fixed cells was performed in the Light Microscopy Facility and Nikon Center of Excellence at the Institute for Applied Life Sciences, University of Massachusetts Amherst with support from the Massachusetts Life Science Center. The authors are grateful to Joe Heitman for his unwavering support throughout this project and to David Booth, Stefano Di Talia, Joe Heitman, and Danny Lew for comments on the manuscript.

## Additional information

### Funding

| Funder | Grant reference number | Author |
| --- | --- | --- |
| National Science Foundation | IOS-1915750 | Nicolas E Buchler |
| National Science Foundation | IOS-1827257 | Lillian K Fritz-Laylin |

The funders had no role in study design, data collection and interpretation, or the decision to submit the work for publication.

### Author contributions

Edgar M Medina, Conceptualization, Investigation, Methodology; Kristyn A Robinson, Kimberly Bellingham-Johnstun, Investigation; Giuseppe Ianiri, Resources; Caroline Laplante, Supervision, Investigation; Lillian K Fritz-Laylin, Supervision, Funding acquisition, Investigation; Nicolas E Buchler, Conceptualization, Supervision, Funding acquisition, Investigation

### Author ORCIDs

Edgar M Medina ⓘ https://orcid.org/0000-0002-5518-5933
Kimberly Bellingham-Johnstun ⓘ https://orcid.org/0000-0001-5608-4010
Giuseppe Ianiri ⓘ http://orcid.org/0000-0002-3278-8678
Nicolas E Buchler ⓘ https://orcid.org/0000-0003-3940-3432

### Decision letter and Author response

Decision letter https://doi.org/10.7554/eLife.52741.sa1
Author response https://doi.org/10.7554/eLife.52741.sa2

## Additional files

### Supplementary files

• Transparent reporting form

### Data availability

All data generated or analysed during this study are included in the manuscript and supporting files.

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

## Appendix 1

## Plasmid construction

We initially built base plasmids and later cloned parts into the T-DNA of the *Agrobacterium* pGI3 plasmid. pGI3 is a binary plasmid derived from pPZP201-BK (KanR) (**Covert et al., 2001**) and pRS426, which contains the *Saccharomyces cerevisiae* 2μ origin of replication and the *URA3* selectable marker. This let us screen gene expression and GFP fluorescence from pGI3EM plasmids transformed into *Saccharomyces*. We identified *H2A* (SPPG_02344), *H2B* (SPPG_02345), and *HSP70* (SPPG_04820) genes and promoters from FungiDB (RRID:SCR_006013) by blasting human homologs against the *Spun* genome. The promoter DNA was cloned from *Spizellomyces* genomic DNA (gDNA) isolated using the methods described in Materials and Methods. Coding DNA was cloned from a cDNA library of *Spizellomyces* transcripts. RNA was extracted from a mixed population *Spizellomyces* zoospores and sporangia using Qiagen RNeasy Plant Mini Kit following manufacturer procedures and RLT buffer. cDNA was synthesized using Thermo Scientific Maxima H Minus Strand cDNA synthesis Kit with dsDNase (REF:K1682) following manufacturer instructions with oligo (dT)$_{18}$ primers.

## Base plasmids

- pEM01 (*CMVpr-hph-ScADH1ter*) was constructed by digesting pNB780 (a pRS406 plasmid) with SacI-HF and BglII, isolating the large backbone fragment, and then assembling CMVpr (primers CMV_F and CMV_R), hph (primers Hyg_F and Hyg_R) and ADH1 terminator (primers ADH1t_F and ADH1t_R) in a single Gibson reaction following manufacturer instructions (NEB, E2611S). CMVpr was obtained by PCR from pNB419 (pAB1T7 = CMVpr-TetR-GFP-VP16), Hygromycin resistance gene (hph) from pRS306H, and *Saccharomyces cerevisiae* ADH1 terminator from pNB780. The final plasmid was verified by analytical restriction digest and Sanger sequencing (primers pRS-up and ADH1t-dn). Plasmid is distributed by Addgene as RRID:Addgene_135414.
- pEM03 (*Hsp70pr-hph-ScADH1ter*) was constructed by digesting pEM01 with SacI and PacI, isolating the large backbone fragment, and then inserting Hsp70pr using Gibson cloning. Hsp70pr was obtained by PCR from *Spizellomyces* gDNA using primers HSP70_F and HSP70_R. The final plasmid was verified by analytical restriction digest and Sanger sequencing (primers pRS-up and ADH1t-dn). Plasmid is distributed by Addgene as RRID:Addgene_135417.
- pEM09 (*Hsp70pr-hph-GFP-ScADH1ter*) was constructed by digesting pEM03 with PacI and AscI, isolating the large backbone fragment, and then inserting hph-GFP using Gibson cloning. Hygromycin resistance gene (hph) was obtained by PCR from pFA6-GFP(S65T):hph using primers HygR_F and HygR_R, whereas GFP(S65T) was amplified from the same plasmid using primers GFP_F and GFP_R. The final plasmid was verified by analytical restriction digest and Sanger sequencing (primers pRS-up and ADH1t-dn). Plasmid is distributed by Addgene as RRID:Addgene_135479.

## *Agrobacterium* plasmids

- pGI3EM09 (*Hsp70pr-hph-GFP-ScADH1ter*) was constructed by digesting pGI3 with HindIII and EcoI, isolating the large backbone fragment, and then inserting Hsp70pr-hph-GFP in the multiple cloning site between the T-DNA LB and RB borders of pGI3 using SLIC cloning (**Li and Elledge, 2007**; **Li and Elledge, 2012**). Hsp70pr-hph-GFP was obtained by PCR of pEM09 using primers GI3EM9IIF and GI3EM9IIR. The final plasmid was verified by analytical restriction digest and Sanger sequencing (Standard primers M13F and M13R). Plasmid is distributed by Addgene as RRID:Addgene_135482.
- pGI3EM11 (*H2Bpr-hph-GFP-ScADH1ter*) was constructed by digesting pGI3EM09 with HindIII and PacI, isolating the large backbone fragment, and then inserting H2Bpr using SLIC cloning. H2Bpr was PCR amplified from *Spizellomyces* gDNA using primers prH2B_F and prH2B_R. The final plasmid was verified by analytical restriction digest and Sanger

sequencing (Standard primers M13F and M13R). Plasmid is distributed by Addgene as RRID: Addgene_135483.

- pGI3EM18 (*H2Bpr-hph-tdTomato-ScADH1ter*) was constructed by digesting pGI3EM11 with HindIII and AscI, isolating the large backbone fragment, and then re-inserting PCR amplicon of H2Bpr-hph with an extra BamHI site 3′ (primers prH2B_F and GI3EM11up) and tdTomato using SLIC cloning. TdTomato was obtained by PCR of pKT356 (pFA6a-tdTomato:SpHIS5; gift from Daniel J. Lew) using primers HygtdTom_F and AdhtdTom_R. The final plasmid was verified by analytical restriction digest and Sanger sequencing (Standard primers M13F and M13R). Plasmid is distributed by Addgene as RRID:Addgene_135484.

- pGI3EM29 (*H2Bpr-hph-mCitrine-ScADH1ter*) was constructed by digesting pGI3EM18 with BamHI and AscI, isolating the large backbone fragment, and then inserting mCitrine using SLIC cloning. mCitrine was obtained by PCR of mCitrine-PCNA-19-SV40NLS-4 (Addgene plasmid # 56564), a plasmid created by the Davidson lab (*Markwardt et al., 2011*). The final plasmid was verified by analytical restriction digest and Sanger sequencing (primers M13 and H2BprF2). Plasmid is distributed by Addgene as RRID:Addgene_135489.

- pGI3EM30 (*H2Bpr-hph-mClover3-ScADH1ter*) was constructed by digesting pGI3EM18 with BamHI and AscI, isolating the large backbone fragment, and then inserting mClover3 using SLIC cloning. mClover3 was obtained by PCR of pKK-mClover3-TEV (Addgene plasmid # 105778), a plasmid created by the Dziembowski lab (*Szczesny et al., 2018*). The final plasmid was verified by analytical restriction digest and Sanger sequencing (primer H2BprF2). Plasmid is distributed by Addgene as RRID:Addgene_135490.

- pGI3EM31 (*H2Bpr-hph-mCerulean3-ScADH1ter*) was constructed by digesting pGI3EM18 with BamHI and AscI, isolating the large backbone fragment, and then inserting mCerulean3 using SLIC cloning. mCerulean3 was obtained by PCR of mCerulean3-N1 (Addgene plasmid # 54730), a plasmid created by the Davidson lab (*Markwardt et al., 2011*). The final plasmid was verified by analytical restriction digest and Sanger sequencing (Standard primers M13F and M13R). Plasmid is distributed by Addgene as RRID:Addgene_135491.

- pGI3EM20B (*H2Bpr-H2B-tdTomato-ScADH1ter:H2Apr-hph-ScADH1ter*) took advantage of a divergent *Spizellomyces* H2A (SPPG_02344) and H2B (SPPG_02345) promoter to express an H2B-tdTomato fusion in one direction (H2B promoter) and hph in the other direction (H2A promoter). The shared promoter region of H2B and H2A corresponds to 217 bp, while the 5′UTR of H2A is 118 bp and H2B is 66 bp for a combined total of 401 bp. This plasmid contained the H2B gene with introns. pGI3EM20B was constructed by digesting pGI3EM18 with HindIII and BamHI, isolating the large backbone fragment, and then assembling hph (from pEM03, primers AdhHygF and AdhHygR), H2A/H2Bpr (primers H2B2D_F and H2B2D_R) and the H2B gene (primers H2Bgen_F and H2Bgen_R) using a four piece SLIC assembly. The final plasmid was verified by analytical restriction digest and Sanger sequencing (Standard primers M13F, M13R and primer H2BprF1).

- pGI3EM20C (*H2Bpr-H2B-tdTomato-ScADH1ter:H2Apr-hph-ScADH1ter*) is a version of pGI3EM20B, where the intron of H2B has been removed. pGI3EM20C was constructed by digesting pGI3EM20B with SalI and BamHI, isolating the large backbone fragment, amplifying the coding version of H2B from *Spizellomyces* cDNA using the same primers (primers H2Bgen_F and H2Bgen_R) followed by SLIC cloning. The final plasmid was verified by analytical restriction digest and Sanger sequencing (Standard primers M13F, M13R, and primer H2BprF1). Plasmid is distributed by Addgene as RRID:Addgene_135487.

- pGI3EM22C (*H2Bpr-Lifeact-tdTomato-ScADH1ter:H2Apr-hph-ScADH1ter*) was constructed by digesting pGI3EM20C with SalI and BamHI, isolating the large backbone fragment, and then inserting LifeAct using standard ligation cloning. LifeAct is a 17 amino acid peptide that binds specifically to polymerized actin in a wide variety of cell types, such as actin patches and cables in yeast and actin-filled protrusions of crawling animal cells (*Riedl et al., 2008*; *Dudin et al., 2019*. LifeAct tag was synthesized as a 300 bp gene fragment through Twist Bioscience in which LifeAct-G(4)S was flanked upstream by SalI restriction site and H2Bpr 5′UTR and downstream by BamHI restriction site and part of the 5′ tdTomato sequence. This gene fragment was amplified by PCR, digested by SalI and BamHI and then ligated into pGI3EM20C backbone fragment. The final plasmid was verified by analytical restriction digest and Sanger sequencing (primer H2BprF2). Plasmid is distributed by Addgene as RRID: Addgene_135488.

