## [Decision Letter]

**Acceptance summary:**

This manuscript reports the first description of genetic transformation of a species of chytrid fungi, a group that contains the pathogens responsible for the large-scale die-offs of amphibian species. The authors demonstrate the use of *Agrobacterium*-mediated transformation to isolate stable transformants of *Spizellomyces punctatus*. This work is likely to be the starting point for work on the molecular biology of chytrids and has potential to establish this chytrid as a comparative model for understanding the evolution of fungal and animal multicellularity and cell biology.

**Decision letter after peer review:**

Thank you for submitting your article "Genetic transformation of *Spizellomyces punctatus* sheds light on the evolution of animal and fungal traits" for consideration by *eLife*. Your article has been reviewed by Ian Baldwin as the Senior Editor, a Reviewing Editor, and two reviewers. The following individuals involved in review of your submission have agreed to reveal their identity: Tim Stearns (Reviewer #2).

The reviewers have discussed the reviews with one another and the Reviewing Editor has drafted this decision to help you prepare a revised submission.

Summary:

This manuscript reports the first description of genetic transformation of a species of chytrid fungi. This is an important technical advance as this fungal group includes *B. dendrobatidis* and *B. salamandrivorans*, two species responsible for large-scale die-off of amphibian species. Although chytrid fungi have been known for more than a century, there has been relatively little genetics and molecular biology done with them, aside from genome sequencing and transcriptomics. The authors demonstrate the use of *Agrobacterium*-mediated transformation to isolate stable transformants of *Spizellomyces punctatus* (Sp), a species that they have previously used as a tool for understanding the evolution of cell cycle control pathways. This work is likely to be the starting point for much more work on the biology of chytrids and the expansion of the molecular genetic manipulations to other chytrid species. In addition, this work has potential to establish this chytrid as a comparative model for understanding the evolution of fungal and animal multicellularity and cell biology.

Given that this is a "Tools and Resources" article, we encourage the authors to revise the title to one that emphasizes the tool they have developed and its potential for driving discovery.

Essential revisions:

1) Given that this is a "Tools and Resources" article with potential to help the community perform experiments in chytrids, we recommend that the authors make strains and plasmids (with a full sequence and map) generally available. Deposition into Addgene or other comparable databases would fulfil this request.

2) While the reviewers appreciate the use of a good, old-fashioned southern to show integration, we would like to see more commentary and/or more data that indicate where sites of integration are. Where are the different transgenes inserted? Are insertion sites based on homology or randomly integrated? Based on insertion sites what is the prospect for performing reverse genetics, not just transgenesis?

3) Along similar lines, southern blotting is used to show that the stable transformants are likely to be single integration events into different loci in the Sp genome. Since the different clones all have different integration sites, the authors should be able to say something about variability of expression of the transformed fluorescent protein gene, based on fluorescence intensity. This is important information, as it would help to understand the range of expression one might expect in clones from a transformation experiment, based on position effect of the random integration site in the genome.

4) Figure 1C should be better labeled. What is (+)? What is the size of the integrating plasmid? Knowing the size would help the reader better evaluate that the larger sized fragments on the southern are true integration sites. Related to this topic, what are the regions surrounding the expression cassette in Figure 2—figure supplement 3? What does RB and LB stand for?

5) In Figure 5A, what explains the difference in LifeAct versus phalloidin localization in the sporangium? (Right most set of panels, green versus magenta). The authors skip over that explanation.

6) The authors describe the use of *Agrobacterium* for transformation in the beginning of the Results section. Since the central advance in this work is the transformation of Sp, and there is likely to be strong interest in the community in transforming other chytrids, it would be helpful to know what other methods of transformation were attempted, presumably without success.

7) There is little information about the promoter and gene segments used in the transformation plasmids. The primer sequences used to amplify them are given, and it would be possible to derive the information from them and the Sp genome sequence, but the reader should not be left having to do that. This is particularly important for the bidirectional histone promoter that is cloned and used in most their experiments. How big is the promoter fragment? Where does it start and stop with respect to the ATGs of the flanking coding sequences?

8) Figure 2—figure supplement 1B: Why are the three tiny colonies pointed out when there are many more on the HygR-tdTomato plate? Are all of those actually transformants, too?

9) Figure 5B, C: The merged images that are shown in these two panels detract from the presentation as they seem to be done in a way that makes them useless for seeing what is going on. Perhaps the DIC was mistakenly put in the colored channel in the merge rather than the LifeAct image?

---

## [Author Response]

Summary:This manuscript reports the first description of genetic transformation of a species of chytrid fungi. This is an important technical advance as this fungal group includes B. dendrobatidis and B. salamandrivorans, two species responsible for large-scale die-off of amphibian species. Although chytrid fungi have been known for more than a century, there has been relatively little genetics and molecular biology done with them, aside from genome sequencing and transcriptomics. The authors demonstrate the use of Agrobacterium-mediated transformation to isolate stable transformants of Spizellomyces punctatus (Sp), a species that they have previously used as a tool for understanding the evolution of cell cycle control pathways. This work is likely to be the starting point for much more work on the biology of chytrids and the expansion of the molecular genetic manipulations to other chytrid species. In addition, this work has potential to establish this chytrid as a comparative model for understanding the evolution of fungal and animal multicellularity and cell biology.Given that this is a "Tools and Resources" article, we encourage the authors to revise the title to one that emphasizes the tool they have developed and its potential for driving discovery.

We revised our title as suggested.

Essential revisions:1) Given that this is a "Tools and Resources" article with potential to help the community perform experiments in chytrids, we recommend that the authors make strains and plasmids (with a full sequence and map) generally available. Deposition into Addgene or other comparable databases would fulfil this request.

We have deposited plasmids and sequence maps to Addgene. All chytrid strains (listed in new Table 3) will be provided by the Buchler lab upon request. We state this information in Methods and Materials.

2) While the reviewers appreciate the use of a good, old-fashioned southern to show integration, we would like to see more commentary and/or more data that indicate where sites of integration are. Where are the different transgenes inserted?

To directly address this question, we performed inverse PCR on the independent transformants from Figure 2 to map where each T-DNA is inserted into the *Spizellomyces* genome. Briefly, genomic DNA was digested with a restriction enzyme followed by ligation to create circularized DNA. Primers that anneal inside the T-DNA close to the left border (LB) and amplify outwards were used for PCR. Positive products were sequenced to recover the junction between the LB and genomic DNA, thus identifying the insertion sites in the genome. Our results (Figure 2—figure supplement 4) show that each transformant is a T-DNA integration event in a different genomic locus, as would be expected from *Agrobacterium*-mediated transformation.

EM20C-1: The T-DNA is integrated in the proximal upstream region (58 bp) of the gene SPPG_04375, a member of the M48 peptidase family.

EM20C-2: The T-DNA is integrated in the proximal upstream region (105 bp) of the gene SPPG_03425, a member of the Usp-like/ adenine nucleotide α-hydrolase family.

EM20C-3: The T-DNA is integrated in the proximal upstream region (189 bp) of the gene SPPG_02523, a PHO-like cyclin. Inverse PCR with EcoRI suggests that LB is closest to SPPG_02523, whereas inverse PCR with HindIII suggests an inverted orientation of the T-DNA in the same locus. One possibility is an irregular integration of T-DNA into the genomic locus. Resolving the architecture of this T-DNA integration through additional inverse PCR or genome sequencing seems beyond the scope of the manuscript because EM20C-3 was not used for downstream experiments.

EM20C-4: The T-DNA is integrated in the distal upstream region (845 bp) of the gene SPPG_08788, a hypothetical protein with no clear conserved protein domains.

Are insertion sites based on homology or randomly integrated? Based on insertion sites what is the prospect for performing reverse genetics, not just transgenesis?

*Agrobacterium* T-DNA is often randomly integrated into the host chromosome as a single copy and, if inserted within a gene, it will disrupt gene function and act as a mutagen (Gelvin, 2017). Random insertional mutagenesis by *Agrobacterium* has been exploited for forward genetics in plants and fungi (Alonso et al. 2003; Michielse et al., 2005; Idnurm et al., 2017). In fungi, *Agrobacterium*-mediated transformation has also been used for targeted gene knock-outs by including 1–1.5 kb long, flanking region of homology in the T-DNA to the desired target gene (Frandsen et al., 2012). The feasibility of this approach relies on the rate of homologous recombination, which is in general high in the majority of Fungi. The low efficiency of homologous recombination in other organisms, such as *Arabidopsis*, has been improved by creating doublestrand breaks in target genes using CRISPR/Cas9.

Although beyond the scope of the current manuscript, the prospect for reverse genetics in *Spizellomyces* and other chytrid fungi looks promising. We plan to use T-DNA plasmids with longflanking homology for targeted gene knock-out/knock-in, and concomitantly create T-DNA plasmids that express Cas9 and sgRNA.

We have incorporated parts of this discussion in the manuscript.

3) Along similar lines, southern blotting is used to show that the stable transformants are likely to be single integration events into different loci in the Sp genome. Since the different clones all have different integration sites, the authors should be able to say something about variability of expression of the transformed fluorescent protein gene, based on fluorescence intensity. This is important information, as it would help to understand the range of expression one might expect in clones from a transformation experiment, based on position effect of the random integration site in the genome.

Indeed, as observed in *Arabidopsis*, we might expect variation in gene expression based on the genomic locus of integration of the T-DNA fragment. To quantify this variation, we measured H2BtdTomato expression in our EM20C transformants in Figure 2 using flow cytometry. Briefly, we grew different transformants and wild type (negative control), harvested zoospores, and measured the tdTomato fluorescence distribution. The data (Figure 2—figure supplement 4) show that H2B-tdTomato expression is identical in 3 out of the 4 transformants, despite the different sites of genomic integration. The exception is EM20C-1, which exhibits bimodal gene expression: the top mode is identical to the other transformants, but the bottom mode is half the intensity. The bimodality is not due to a mixed population with variable T-DNA copy number and/or genomic integrations because Southern blot and inverse PCR data indicate a single copy integration event in EM20C-1. Epigenetic suppression of T-DNA has been observed in *Arabidopsis*, and bimodal H2B-tdTomato expression in EM20C-1 might arise from epigenetic regulation of the transgene in its specific genomic locus.

Our previous time lapse movies (Figure 2 and Figure 3) were done with transformant EM20C-1. To check whether the location and/or expression level of H2B-tdTomato transgene might affect previous conclusions regarding chytrid development and cell cycle, we made replicate time-lapse fluorescence movies of EM20C-1 and EM20C-2 (Figure 2—figure supplement 5). We scored the timing of germ tube formation, mitosis (i.e. nuclear division), and sporangium bursting in single cells. For either strain, one replicate initiated the developmental program earlier than the other replicate by 1-2 hours. This systematic time shift between replicates, which propagates to downstream events, is likely due to a (currently) unknown and uncontrolled factor that regulates early developmental events. Despite this systematic time shift between replicates, both EM20C1 and EM20C-2 formed germ tubes at 1-3 hours, the first mitosis occurred between 8-12 hours, and the average cell cycle period in early development was ~150 minutes. This cell cycle period is identical to that of EM20C-1, previously measured by confocal fluorescence time lapse (Figure 3). Despite the overall similarity between transformants, there is one significant difference:

EM20C-2 had a shorter developmental cycle than EM20C-1 (bursting at 20 – 25 hours versus 25 – 30 hours). This stemmed from a combination of shorter cell cycle periods in late development and fewer mitotic cycles before bursting.

We have incorporated parts of this discussion in the manuscript. We also modified the timing of developmental events in the caption of Figure 1 to mirror EM20C-1 and EM20C-2 time-lapse data from Figure 2—figure supplement 5.

4) Figure 1C should be better labeled. What is (+)? What is the size of the integrating plasmid? Knowing the size would help the reader better evaluate that the larger sized fragments on the southern are true integration sites. Related to this topic, what are the regions surrounding the expression cassette in Figure 2—figure supplement 3? What does RB and LB stand for?

Figure 2C: The (+) refers to the positive control (GI3EM20C plasmid). We now describe this control in the caption.

Figure 2—figure supplement 3: RB and LB are the "right border" and "left border" of the T-DNA plasmid. These sequences define the T-DNA fragment that is excised and injected into the host cell by the *Agrobacterium* virulence machinery upon induction. The total size of the T-DNA (left border, LB, to right border, RB) is 4280 bp. We now include this information in the figure legend.

5) In Figure 5A, what explains the difference in LifeAct versus phalloidin localization in the sporangium? (Right most set of panels, green versus magenta). The authors skip over that explanation.

The minor deviations between LifeAct and phalloidin can be explained by intrinsic biases of the probes. All live-cell probes of filamentous actin (e.g. LifeAct, F-tractin, Utrophin actin binding domain) have different biases in their patterns of F-actin localization and dynamics (Belin et al., 2015). None of them can fully recapitulate the patterns observed with phalloidin.

We now clarify this in the text.

6) The authors describe the use of Agrobacterium for transformation in the beginning of the Results section. Since the central advance in this work is the transformation of Sp, and there is likely to be strong interest in the community in transforming other chytrids, it would be helpful to know what other methods of transformation were attempted, presumably without success.

We initially tried zoospore electroporation using a protocol developed in zoosporic protists, such as *Phytophthora* (Ah-Fong et al., 2018). This was unsuccessful and we turned to *Agrobacterium-*mediated transformation because it has worked in other fungi and because we had access to the pGI3-based plasmids and expertise of Giuseppe Ianiri (co-author and former postdoc at Duke University in the lab of Joe Heitman). This worked and, thus, we did not continue exploring other methods of transformation (e.g. protoplast lithium acetate, biolistics, etc.).

We have added a brief description in the Materials and methods section.

7) There is little information about the promoter and gene segments used in the transformation plasmids. The primer sequences used to amplify them are given, and it would be possible to derive the information from them and the Sp genome sequence, but the reader should not be left having to do that. This is particularly important for the bidirectional histone promoter that is cloned and used in most their experiments. How big is the promoter fragment? Where does it start and stop with respect to the ATGs of the flanking coding sequences?

The shared/combined promoter region of H2B and H2A is 217 bp. We also included endogenous 5’UTR regions of the H2A (118 bp) and H2B (66bp), resulting in a total of 401 bp. This cloned bidirectional "promoter region" is followed by restriction sites and the start codons of the corresponding proteins of interest (hph or H2B CDS). This is schematically depicted in Figure 2—figure supplement 3A.

We also include this specific information in the Appendix and updated Figure 2—figure supplement 3A.

8) Figure 2—figure supplement 1B: Why are the three tiny colonies pointed out when there are many more on the HygR-tdTomato plate? Are all of those actually transformants, too?

We highlighted these colonies because they are difficult to see without a visual cue, unlike the hph-tdTomato plate. We did not test all hph-tdTomato transformants with PCR and fluorescence microscopy, but we suspect that most of them would be true transformants. The frequency of hygromycin resistant colonies in hph-GFP and hph-tdTomato was consistently higher than the no plasmid control (i.e. no colonies).

We now show photos of the transformation plates using inverted contrast to better highlight the colonies.

9) Figure 5B, C: The merged images that are shown in these two panels detract from the presentation as they seem to be done in a way that makes them useless for seeing what is going on. Perhaps the DIC was mistakenly put in the colored channel in the merge rather than the LifeAct image?

When multiple channels are shown for an image (including DIC) it is best practice to include a merged image of the independent channels (Lee and Kitaoka, 2018). If the journal agrees, we would prefer to keep the merges.